# Disparate substrates for head gaze following and face perception in the monkey superior temporal sulcus

Karolina Marciniak*, Artin Atabaki, Peter W Dicke, Peter Thier*

Department of Cognitive Neurology, Hertie Institute for Clinical Brain Research, University of Tuebingen, Tuebingen, Germany

**Abstract** Primates use gaze cues to follow peer gaze to an object of joint attention. Gaze following of monkeys is largely determined by head or face orientation. We used fMRI in rhesus monkeys to identify brain regions underlying head gaze following and to assess their relationship to the 'face patch' system, the latter being the likely source of information on face orientation. We trained monkeys to locate targets by either following head gaze or using a learned association of face identity with the same targets. Head gaze following activated a distinct region in the posterior STS, close to-albeit not overlapping with-the medial face patch delineated by passive viewing of faces. This 'gaze following patch' may be the substrate of the geometrical calculations needed to translate information on head orientation from the face patches into precise shifts of attention, taking the spatial relationship of the two interacting agents into account.

*For correspondence: marciniak.
kar@gmail.com (KM); thier@
uni-tuebingen.de (PT)

Competing interests: The authors declare that no competing interests exist.

## Introduction

Successful social interactions require understanding of peer dispositions, desires, beliefs and intentions. A major step in developing this *theory of* (the other one's) *mind* (TOM) is our ability to shift our attention to the same location and/or object the other one is interested in, that is to establish *joint attention* (*Baron-Cohen, 1995*). By associating our own object-related aspirations and intentions with the object, we may arrive at a viable theory of the other one's mind. In order to shift our attention to the right place or object, we rely on spatial information provided by peer body such as the direction of a pointing finger or the orientation of the head and the shoulder girdle. In case of humans, arguably the most important bodily cue, reliably providing extremely precise spatial information (*Bock et al., 2008*), is peer eye gaze, a cue that can be easily retrieved even from quite some distance because of the high contrast border between the eye's dark center and periphery. The eyes of macaque monkeys and many other nonhuman primates (NHP) lack comparable contrasts (*Kobayashi and Kohshima, 1997*), which is why eye gaze seems to be of little importance in this group (*Lorincz et al., 1999*). While monkeys seem to lack a full-fledged TOM (*Anderson et al., 1996*) (*Flombaum and Santos, 2005* for a different view), they are nevertheless able to establish joint attention with conspecifics, largely relying on head gaze, that is the orientation of peer head (=face). Importantly, not only human but also monkey gaze following seems to be geometric: the observer identifies peer focus of attention by following his/her gaze towards the object of interest (*Emery et al., 1997*; *Emery, 2000*). In other words, rather than simply using the directional information provided by a gaze cue to shift attention out from the center until an object of potential interest is encountered, geometrical gaze following implies that a gaze vector is defined which is used to search for the object of interest (*Butterworth and Jarrett, 1991*). Another important feature shared by the gaze following of monkeys and man is the dependence on social context. For instance, human observers tend to prefer gaze cues of those whom they feel close to *Liuzza et al. (2011)*, while monkeys are particularly eager to follow the gaze

**eLife digest** Gaze following—working out where someone else is looking, and then switching your attention to that position—is an important part of social behavior and learning. Additionally, it is thought to be an important step towards recognizing that others have a mind of their own. Humans mostly use eye position to work out the 'gaze direction' of someone else, whereas non-human primates rely instead on the orientation of the face. However, the neural circuits that control gaze following are thought to be similar in both.

Gaze following is a complex process that requires the brain to process a lot of different information. A face must be recognized, and its orientation worked out. A series of complex geometrical calculations must then be performed to work out the direction of the gaze, and how this relates to the position of the observer. Finally, the object of interest needs to be recognized and the attention of the observer focused on it.

In the monkey brain, there are six interconnected areas called face patch regions that respond when a monkey is shown a face. However, researchers do not understand how monkeys translate the information about face orientation gathered by these regions into information about where to look during gaze following.

Marciniak et al. performed functional magnetic resonance imaging on monkeys to track the flow of blood to different regions of the brain—the higher the blood flow, the more that area of the brain is working. To identify the location of their face patch regions, the monkeys first looked at faces. When the monkeys then performed a gaze following task, a region of the brain close to—but not overlapping—the face patches was activated. Marciniak et al. suggest this is the 'gaze following patch' where the brain performs the demanding calculations to translate face orientation into a position to look at.

As gaze following is important in social interactions, understanding the neural circuits behind it could help us understand social disorders.

of higher status conspecifics (*Shepherd et al., 2006*). The availability of geometric gaze following in monkeys and man and the modulatory influence of context supports the idea that monkey and human gaze following may actually be closely related, sharing homologous substrates, although the choice of the relevant social cue—eye vs head—differs. In any case, perceiving peer eye or head gaze and converting it into a gaze vector is only a first step in a sequence of demanding computations that ultimately lead to the establishment of joint attention. This is a consequence of the fact that objects of interest may lie anywhere relative to the demonstrator and the observer. Only if the object were midway between the two, joint attention could be established by simply mirroring the demonstrator's gaze vector. However, as this specific location will be an exception rather than the rule, the object position will have to be transformed from a demonstrator-centered frame of reference (FOR) into an observer-centered FOR before a successful shift of attention can be programmed.

Previous work on gaze following suggests that the computational steps leading from the extraction of eye and/or head direction to shifts of attention are based on a distributed network of areas located in the superior temporal sulcus and the posterior parietal cortex. For instance, single unit recordings from monkey area LIP suggest that it is the major substrate of the shifts of attention which are prompted by social cues providing spatial information (*Shepherd et al., 2009*) as well as by non-social spatial cues (*Bisley et al., 2011*). On the other hand, BOLD imaging studies of human cortex have consistently singled out an area in the posterior superior temporal sulcus (pSTS) specifically activated by the processing of eye gaze cues leading to subsequent shifts of attention (*Materna et al., 2008a*). BOLD imaging is also able to delineate a patch of cortex in the monkey STS activated by head gaze following, arguably offering similar functionality as the human pSTS region and perhaps even being homologous (*Kamphuis et al., 2009*). Yet, what this specific functionality might be remains unclear. An obvious possibility is that this 'gaze following patch' may contribute to extracting relevant facial features. This possibility is supported by the fact that previous work on face processing has established a set of six disparate but interconnected 'face patches' distributed along the rostro-caudal extent of the monkey STS, that is found in the same general region as the gaze following patch (*Tsao et al., 2003*, *2008*). As previous work on face processing has suggested that these face patches form a

hierarchically organized network with functionally specialized nodes (*Moeller et al., 2008*; *Freiwald and Tsao, 2010*), we reasoned that one of these nodes might actually correspond to the gaze following patch. Using fMRI to delineate the passive face patch network and the gaze following patch in the same rhesus monkeys, we here show that, surprisingly, this expectation is not met. Rather than being part of the face patch network, the gaze following patch remains outside the network, albeit close to one of its nodes, the medial face patch.

## Results

### Behavioral results

In the first experiment, the two monkey subjects ('observers') were exposed to the portraits of fpur monkey individuals turning their head at one out of four different targets arranged along the horizontal. The two observers were instructed to either use head gaze orientation to identify the spatial target (gaze following) or, alternatively, to exploit learned associations between the identity of the seen faces and the individual targets (identity-matching condition) while ignoring head gaze (*Figure 1A,B*). The target choice was indicated by precise saccades to one of the four targets (*Figure 1C*). In other words, the two conditions were the same in terms of the visual information provided by the presented portraits as well as the motor behavior the visual stimuli prompted. However, they differed with regard to the facial cues used to prompt shifts of attention. We compared the accuracy of behavioral responses as measured by the percentages of correct choices for the gaze following and the identity-matching task, using data sets underlying the fMRI analysis. The accuracy levels were not significantly different in the two tasks (*Figure 1D*). Saccade latencies were also not different (*Figure 1E*).

The two control tasks used to rule out that monkeys might have managed to circumvent true gaze following in the gaze following task by resorting to learned associations between the position of the portraits on the screen and the spatial (or ordinal) position of the targets, were carried out only outside the scanner. We assigned responses to three possible categories, the first one consistent with gaze following, the second one comprising responses reflecting a workaround strategy (control experiment 1: learned spatial associations; control experiment 2: learned order associations) and the third one, neither consistent with category 1 nor 2 (*Figure 2A,B*; 'Materials and methods'). In each and every case the number of responses in the gaze following category by far surpassed the numbers in the two other categories (*Figure 2C,D*). The dominance of gaze following was statistically highly significant (*Figure 2C,D*). These behavioral results clearly demonstrated that the observers M1 and M2 followed the gaze of the monkey portrait in the gaze following task. To solve the identity-matching task, the two monkeys obviously used individually adjusted strategies to identify the correct target. Depending on the behavioral context, they relied primarily on associations of individual identities with absolute spatial position or on relative spatial order (*Figure 2—figure supplements 1–4*).

### Comparison of BOLD responses to gaze following and identity matching (Experiment 1)

To identify brain regions specifically activated when the observers relied on gaze orientation, we looked for voxels showing significantly larger BOLD responses to gaze following than to identity matching. Analyzing the whole brain of M1 for the 'gaze following > identity matching' BOLD contrast, the only region showing a significant contrast was a patch of voxels (='gaze following (GF) patch') located unilaterally on the lower bank of the right STS, between the interaural line and 2 mm posterior to it (A0-P2), near to the dorsal end of the inferior occipital sulcus (*Figure 3*). We reasoned that the absence of a significant BOLD contrast on the opposite site might have been the consequence of a poorer signal-to-noise ratio (SNR) on the left side, possibly due to the positioning of the eye-tracking camera in front of the left eye. To test it, we performed additional experiments in M1 with a focal coil centered on the posterior left STS, expected to be less affected by the camera. Yet, also with this coil we did not obtain a significant BOLD contrast in the area sampled by the coil, which included the region corresponding to the activated GF patch on the opposite side. Interestingly, despite the fact that the focal coil was centered on the left STS, it allowed us to confirm the right posterior STS BOLD contrast obtained in a whole-brain analysis of the data collected with the bilateral coil (*Figure 3—figure supplement 1*). Finally, also the possibility that the unilateral BOLD pattern might be secondary to differences in the behavior directed at the two sides can be excluded. A comparison of responses to the demonstrator's gaze directed to the left and to the right respectively did not reveal any differences in accuracy ('Materials and methods').

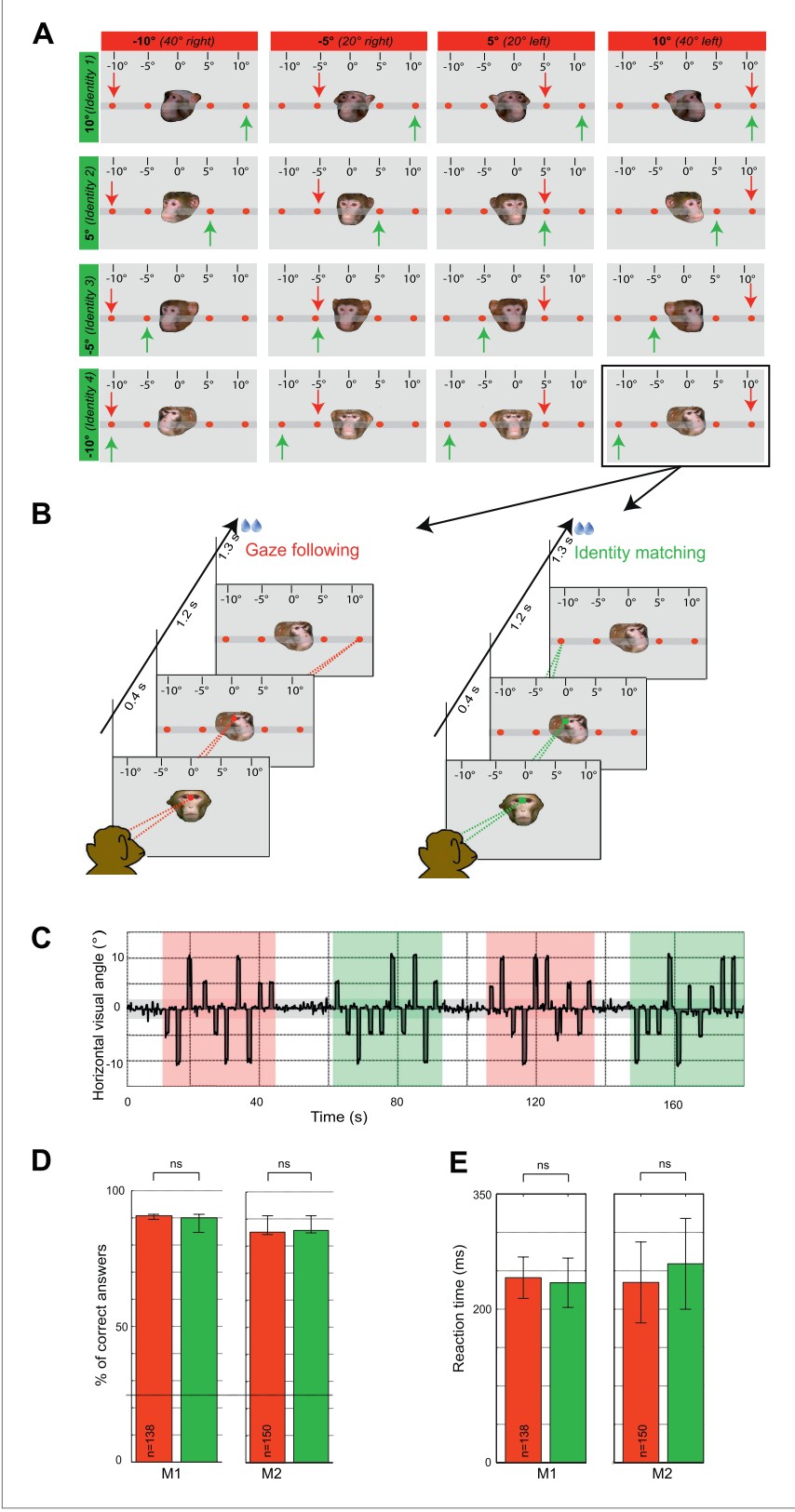

**Figure 1.** Experimental paradigm and behavioral results ('gaze following' paradigm). (**A**) Stimuli. 16 portraits used in the gaze following and identity matching tasks, arranged by the same identity (rows) or head orientations
*Figure 1. Continued on next page*

*Figure 1. Continued*

(columns, demonstrator's head orientation eccentricity indicated in brackets). The arrows point to the correct target dot in gaze following (red) and identity matching task (green). Arrows and the scale with the eccentricity of the target as seen by the observer were not visible during the experiment. Portraits and target bar were presented on an otherwise black background (here shown as gray for better visualization). (**B**) Sequence of events. Exemplary gaze following (left) and identity matching (right) trials. (**C**) Exemplary horizontal eye movements sampled during a typical fMRI run. The gray shaded horizontal area around 0° indicates the limits (±2°) of the fixation window, the red areas indicate gaze following blocks and the green ones identity matching blocks. White areas outline the 'fixation-only' blocks. (**D**) Median percentages of correct answers in gaze following (red) and identity matching blocks (green), pooled separately for each observer (M1: 138 blocks; M2: 150 blocks) in 'gaze following' paradigm. Error bars represent 95% confidence intervals. The difference was not significant (ns, Wilcoxon signed rank test: p=0.67 [M1], p=0.43 [M2]). Dashed line indicates the chance level in each task (25%). (**E**) Mean reaction times in gaze following (red) and identity matching blocks (green), pooled separately for the two observers (M1:138 blocks; M2: 150 blocks) in 'gaze following' paradigm. Error bars represent standard errors. The difference was not significant (ns, paired samples *t* test: p=0.08 [M1]; p=0.22 [M2]).

Since the only significant BOLD activation yielded by the whole-brain analysis of M1 was in the STS, we focused our scanning onto the temporal lobes of M2, using a bilateral and a unilateral coil configuration ('Materials and methods'). By this approach we revealed a significant BOLD contrast for gaze following compared to identity matching in the lower bank of the STS on both sides, around 1 mm anterior to the interaural line (A1), near the dorsal end of the inferior temporal sulcus (*Figure 3*). The GF patch in the right hemisphere of M2 was shifted by 1–2 mm anterior to the coordinates of the GF patch in M1. It is important to emphasize that the patches were singular in both monkeys and located in the same general part of the STS. This strongly suggests that the slight shift is a manifestation of interindividual variability and thus does not question the spatial identity of the GF patch in the two monkeys. However, it is harder to explain the fact that M2 in the left STS had 3 disparate patches of gaze following-associated BOLD. In terms of their location, the posterior-most patch, which exhibited a much stronger peak BOLD signal than the other two, corresponds to the GF patch on the right side in terms of coordinates. On the other hand, the peak BOLD responses of the two more anterior patches located in the left STS, around A4 and A8 respectively, were much weaker, although consistent across the usage of the two different coil systems. We will reserve the term 'GF patch' to the posterior patch, consistently showing gaze following-associated BOLD activity and use the qualifier 'anterior' when discussing the two anterior patches in the left STS of M2.

For the 'identity matching > gaze following' BOLD contrast we reached significance only at a level of p<0.05 (uncorrected). In M1 the activity was found unilaterally on the lower lateral bank of the right STS around 17 mm anterior to the interaural line (A17). In M2 it was bilateral in the medial part of the STS around 22 mm anterior to the interaural line (A22).

## BOLD activation related to the perception of faces (Experiment 2)

Analyzing the whole brain of M1 and focusing on the temporal lobes of M2, we identified a pattern of face-specific BOLD activations consisting of a number of distinct patches. Their coordinates and their spatial layout corresponded to the face patches described by previous work (*Tsao et al., 2003*, *2008*; *Moeller et al., 2008*) and at least partially confirmed by subsequent studies (*Pinsk et al., 2005*; *Bell et al., 2009*; *Ku et al., 2011*): three bilateral anterior face-patches (anterior medial, AM; anterior lateral, AL; anterior fundus, AF), two middle bilateral face-patches (middle lateral, ML and middle fundus, MF) and one bilateral posterior face patch (posterior lateral, PL) (*Figure 4*). The results were consistent across the two monkeys and consistent across the two different receiver coil systems used in M2 (focal unilateral coil vs bilateral coil). The least clear patches were AM, AF and PL, as the BOLD activation for those voxels reached significance only at a level of p<0.05 (uncorrected), whereas the other clusters (ML, MF and AL) reached significance already on a level of p<0.005 (uncorrected).

## The spatial relationship of gaze following-related BOLD activity and the face patch system

To explore the relationship of the GF patch (Experiment 1) to the face patches found in Experiment 2, we projected the BOLD patterns obtained in the two experiments onto coronal sections and onto a

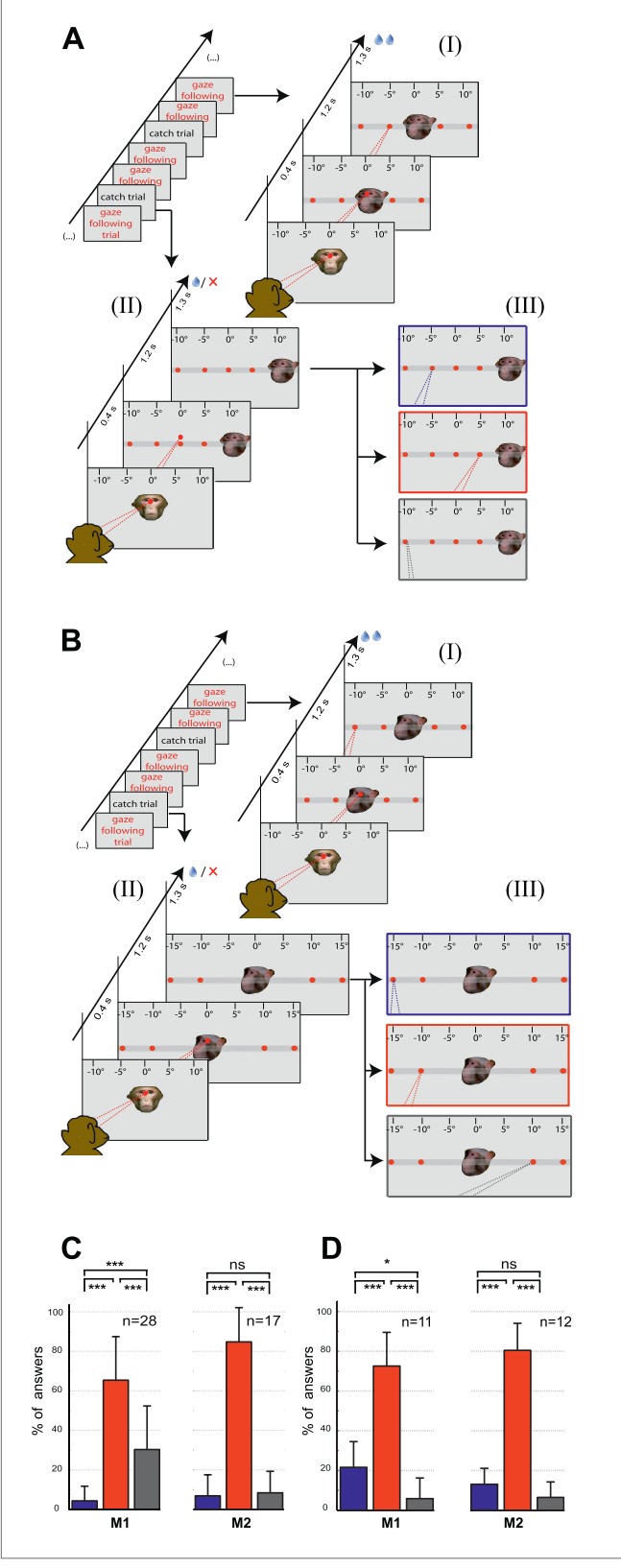

**Figure 2**. Control experiments. (**A**) Testing for learned associations between head orientation and the spatial position of the target. Sequence of normal gaze following trials (I) with catch trials (II) where demonstrator portrait

*Figure 2. Continued on next page*

*Figure 2. Continued*

was shifted horizontally (here by 10°). Subject's responses in the catch trials were later classified into three categories (III): (1) The 'gaze following' category (red outline). (2) The 'learned spatial association' category (blue outline). (3) The 'other' category (gray outline). Dashed lines in the figures indicate the observer's eye gaze. (**B**) Testing for associations between head orientation and the ordinal position of targets. Sequence of normal gaze following trials (I) with catch trials (II) where the 10° eccentricity targets maintained their standard spatial position but changed their ordinal position (II). The responses in catch trials were later classified into three categories (III): (1) The 'gaze following' category (red outline). (2) The 'learned order association' category (blue outline). (3) The 'other' category (gray outline). (**C**) The results of control Experiment 1 (*Figure 2A*). Mean percentages of responses classified as the 'gaze following' category (red column), the 'learned spatial associations' category (blue column) and in the 'other' category (gray column). Both monkeys showed significantly more responses in the 'gaze following' category than in the two other ones (repeated measures 1-way ANOVA, significant effect of the factor 'response category' ($F_{2,54}$ = 51.23, p<0.001 [M1]; $F_{2,32}$ = 127.4, p<0.001 [M2]). (**D**) The results of the control Experiment 2 (*Figure 2B*). Mean percentages of responses classified as the 'gaze following' category (red column), the 'learned order associations' category (blue column) and in the 'other' category (gray column). Both monkeys showed significantly more responses in the 'gaze following' category than in the two other ones (repeated measures 1-way ANOVA, significant effect of the factor 'response category' [$F_{2,20}$ = 47.8, p=0.001 (M1); $F_{2,22}$ = 132.2, p<0.001 (M2)]); In [**C**] and [**D**] post hoc pairwise comparisons [with Bonferroni correction] are indicated with significance levels: ***p<0.001, **p<0.01, *p<0.05, not significant [ns]; n indicates the number of experimental repetitions. Error bars represent standard errors. M1 = monkey 1, M2 = monkey 2.

The following figure supplements are available for figure 2:

**Figure supplement 1**. Design of the control experiments testing for learned associations between facial identity and the spatial position of the target.

**Figure supplement 2**. Design of the control experiment testing for associations between the facial identity and the ordinal position of targets.

**Figure supplement 3**. The results of the control experiment testing for associations between demonstrator's identity and target location (*Figure 2—figure supplement 1*).

**Figure supplement 4**. The results of the control experiment testing for associations between demonstrator's identity and the ordinal position of targets (*Figure 2—figure supplement 2*).

partially unfolded 3D representation of the individual monkey's brain, based on anatomical MRI images. The GF patch in the posterior STS did not overlap with any of the face patches we identified (*Figure 4*). Actually, the GF patch was posterior (with respect to the interaural line) relative to the middle face patches (ML and MF) and anterior to the posterior face patch (PL). Moreover, despite the slight difference between the location of the GF patch in the two monkeys the relative distance between this patch and the middle face patch (ML) matches in the two monkeys. In other words, the middle face patches of the two monkeys showed a comparable spatial offset to the GF patch. Unlike the major (posterior) GF patch, the additional anterior gaze following patches found in the left hemisphere of M2 only, overlapped with the MF face patch in this monkey. Interestingly, the weak anterior patch we had found for the opposite, 'identity matching vs gaze following' contrast, overlapped with anterior face patches: in M1 the unilateral identity patch on the lower bank of the STS around A17 right overlapped with the anterior lateral face patch (AL) in this monkey. In M2 the bilateral identity patches on the medial part of the STS around A22 overlapped with the anterior medial face patch (AM).

## Discussion

We asked monkeys to use head gaze orientation of a portrayed conspecific to identify the spatial target the portrayed monkey was looking at, and to overtly shift attention to the same target. If cued by an alternative instruction, the same subjects exploited learned association between portrait identities and specific target locations to shift attention while ignoring gaze.

Rather than to engage in geometrical head gaze following, the experimental animals might have learned to associate the head orientation of the portrayed monkey with particular target positions, either defined in absolute or in relative terms. We could rule out that the experimental animals relied

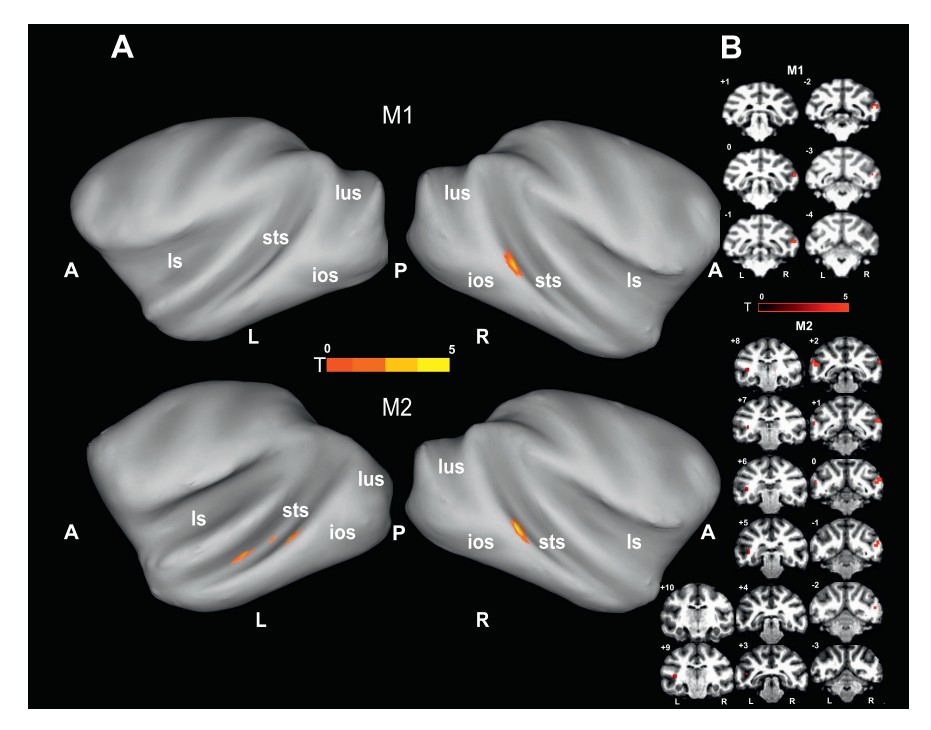

**Figure 3**. 'Gaze following vs identity matching' BOLD contrast. (**A**) Lateral views of the partially inflated hemispheres of monkeys M1 and M2 with significant (p<0.005, uncorrected, 5 contiguous voxels) BOLD 'gaze following vs identity matching' contrasts. A = anterior, P = posterior, L = left, R = right, sts = superior temporal sulcus, ios = inferior occipital sulcus, lus = lunate sulcus, ls = lateral sulcus. (**B**) Coronal sections through the brains of monkeys M1 and M2 with corresponding significant BOLD contrast from (**A**). The color scale bar gives the t-scores indicating the size of significant BOLD contrasts. The numbers in the left corners of each section indicate the distance from the vertical interaural plane of each monkey. L = left, R = right.

The following figure supplements are available for figure 3:

**Figure supplement 1**. 'Gaze following vs identity matching' BOLD contrast evoked in Experiment 1 using a unilateral small coil placed on the left hemisphere of M1.

on a workaround strategy based on such learned associations by presenting catch trials in behavioral control experiments in which spatial targets were presented in new associations with the portraits. In these catch trials, the experimental animals clearly preferred the target defined by head gaze of the portrayed monkey, rather than being misdirected to targets defined by one of the associations with portraits experienced in the training set. Hence, we can be sure that our experimental animals deployed geometrical head gaze following (*Emery et al., 1997*; *Emery, 2000*).

Using gaze direction rather than identity to shift attention was associated with a distinct patch of BOLD activity (GF patch) in a region of the posterior STS, on its lower bank around A1-P1, near the dorsal end of the inferior temporal sulcus, corresponding to cytoarchitectonic area TEO (*Bonin and Bailey, 1947*; *Ungerleider and Desimone, 1986*) and area PITd, the latter defined based on topographic and connectional data (*Felleman and Van Essen, 1991*). The location of the GF patch in this study is more posterior than BOLD activity, interpreted as being gaze following related, in two other monkeys used in a previous study by *Kamphuis et al. (2009)*. In that study, gaze following-related activity was found between A4.4-6.4 in one animal and A0.8-A3.3 in the other one. In other words, the anterior deviation of the location of the gaze following activity in the second monkey of Kamphuis et al. relative to the GF patch in the present study is not much larger than the interindividual differences in the present study (M1:A0-P2, M2: A2-P1). Hence, we may safely assert agreement. The question is if the much more anterior location of gaze following-related BOLD activity in the first monkey studied by Kamphuis et al. is a reflection of more substantial interindividual variability or, alternatively,

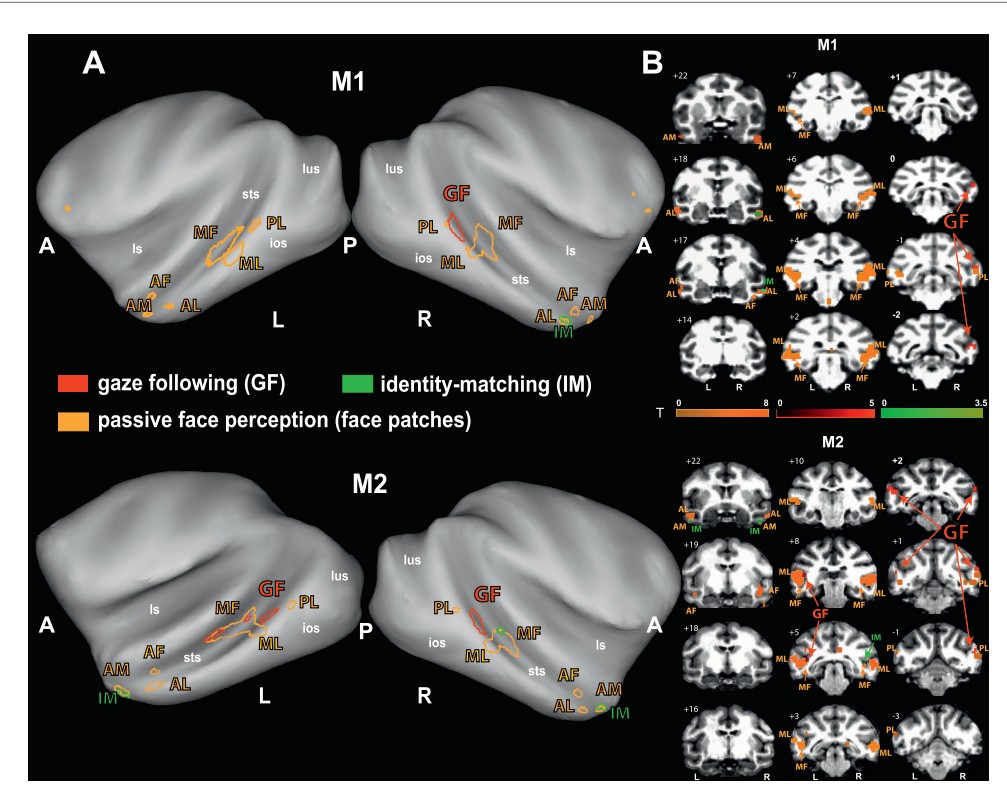

**Figure 4**. Comparison of the patterns of BOLD responses to 'gaze following' and 'identity matching' with the face patch BOLD pattern, delineated by the passive viewing of faces. (**A**) Lateral views of the partially inflated hemispheres of monkeys M1 and M2 with borders of significant BOLD responses. Face patches (orange) based on 'faces vs nonfaces' contrast (p<0.05, uncorrected, 5 contiguous voxels) masked with an 'all non-scrambled vs all scrambled' objects' contrast (p<0.05 uncorrected). The red contours: significant BOLD contrasts for the 'gaze following vs identity matching' comparison (p<0.005, uncorrected, 5 contiguous voxels). The green contours: significant BOLD contrasts for the opposite, 'identity matching vs gaze following' comparison (p<0.05, uncorrected, 5 contiguous voxels). A = anterior, P = posterior, L = left, R = right, sts = superior temporal sulcus, ios = inferior occipital sulcus, lus = lunate sulcus, ls = lateral sulcus. (**B**) Coronal sections through the brains of monkeys M1 and M2 with corresponding significant BOLD contrasts from (**A**). The numbers in the left corners indicate the distance from the vertical interaural plane of each monkey (positive values anterior, neg. posterior) (L = left, R = right).

The following figure supplements are available for figure 4:

**Figure supplement 1**. Examples of stimuli used in the 'passive face perception' experiment.

a consequence of a more fundamental difference between this monkey and the others. Actually, a limitation of the Kamphuis et al. study was that because of a lack of behavioral controls the authors could not exclude that the experimental animals might not have resorted to a workaround strategy based on learned associations between portraits and targets. Moreover, the two conditions compared in order to identify gaze following-related BOLD activity involved subtle visual differences. Hence, a possibility is that the pattern evoked in the Kamphuis et al. study may have been influenced by variables other than gaze following and we may speculate that these variables may have played a larger role in the odd monkey.

Whereas in one monkey the GF patch could be delineated bilaterally in the STS in corresponding locations, much to our surprise, the gaze following-related activity in the other monkey was confined to the right hemisphere. Our attempts to lead this unilaterality back to differences in the quality of the MRI signal on the two sides or to differences in the behavior directed at the two sides, failed. Whatever the reason for the unilaterality may be, it seems specific to gaze following as testing the same monkey in a passive face perception task to delineate the face patch network did not reveal any

major hemispheric differences. Previous behavioral studies on gaze following in healthy human subjects and patients with lesions (*Ricciardelli et al., 2002*; *Akiyama et al., 2006a*, *2006b*) supports a specific role of the right hemisphere in gaze following and social cognition in general (*Brancucci et al., 2009* for review). On the other hand, fMRI studies of gaze following in humans have yielded mixed results, with some showing mainly right hemisphere STS activation (*Pelphrey et al., 2003*, *2004*, *2005*; *Calder et al., 2007*; *Laube et al., 2011*) and others bilateral pSTS activation (*Puce et al., 1998*; *Hoffman and Haxby, 2000*; *Materna et al., 2008a*, *2008b*). Hence, it is tempting to see certain analogies between the human data and our monkey findings and to conclude that gaze following in monkeys and man may in principal be bihemispheric, though the substrates on the left side are deployed to individually varying degrees. The idea of true hemispheric differences in monkeys receives further support from the fact that the left STS of monkey M2 not only showed a GF patch with coordinates matching those of the patch in the right hemisphere but in addition two more anterior patches without equivalents in the right hemisphere of this monkey and none of the two hemispheres of monkey M1. Actually, recent work on audio-vocal communication in macaque monkeys, emphasizing an advantage of the left hemisphere (*Heffner and Heffner, 1984*; *Ghazanfar and Hauser, 2001*; *Poremba et al., 2004*), supports the idea that there is some hemispheric specialization in macaque monkeys. In any case, more subjects will have to be studied to decide if the seeming unilaterality in monkey M1 is more than an idiosyncrasy.

Using the same face patch localizer paradigm as used in previous work (*Tsao et al., 2003*; *Moeller et al., 2008*; *Tsao et al., 2008*), we delineated the set of face patches with comparable coordinates and spatial layout. The only difference with respect to the later studies of Tsao et al. (*Moeller et al., 2008*; *Tsao et al., 2008*) was a somewhat weaker and less consistent activation of the anterior and posterior face patches. This difference is most probably a consequence of methodological differences: these later studies (*Moeller et al., 2008*; *Tsao et al., 2008*) relied on MION-based measurements of activity-related changes in blood volume, well-known to be more sensitive by a factor of 3 than the BOLD method deployed by us, using the same 3T scanner (*Leite et al., 2002*). Actually, when relying on BOLD imaging of the monkey face patch system and deploying a comparably high significance threshold, also an earlier study of *Tsao et al. (2003)* identified only a fraction of the face patches which were later demonstrated with MION. In this earlier study the strongest activity was found in the face patches in the fundus and lower bank of the middle STS (corresponding to ML and MF in *Moeller et al., 2008*; *Tsao et al., 2008*) and the face-patch located in rostral TE (corresponding to AL in *Moeller et al., 2008*; *Tsao et al., 2008*). The patch in the STS in area TEO (corresponding to PL in *Moeller et al., 2008*; *Tsao et al., 2008*) was not reliable across different days and other anterior face patches (AF and AM) were not reported. This pattern fits our results. Nevertheless, we clearly identified all the medial and posterior face patches described before (*Moeller et al., 2008*; *Tsao et al., 2008*) which were in the vicinity of our GF patch. This is important as our major finding is the complete separation of the GF patch from any of the neighboring face patches with ML/MF being closest to the GF patch. The face patch system in monkeys is largely bilateral (*Tsao et al., 2003*, *2008*). The fact that unlike face-related activity, the gaze following-related activity was unilateral in one of the two monkeys studied, further supports the notion of two distinct and anatomically separated systems. On the other hand, the two weak BOLD responses observed inconsistently much more anterior in conjunction with gaze following overlapped with the MF face patch. This overlap may suggest that MF may be more important for processing information on facial orientation than on facial identity. Physical proximity does not necessarily imply connectivity and close functional relationship. Yet, the properties of neurons in ML/MF are suggestive of a functional relationship: many face-selective cells are tuned to specific face (=head) orientations (*Freiwald et al., 2009*; *Freiwald and Tsao, 2010*). This is exactly the kind of information head gaze following builds on.

To identify the goal of the other one's gaze in the frame of reference of the observer or, alternatively, in a world-centered frame of reference shared by both agents, the spatial relationship of the two agents and the relationship of potential goal objects relative to the two agents needs to be taken into account also. Hence, it is intriguing to speculate that the GF patch could be the substrate of the geometrical calculations needed to establish this goal representation, to this end adding the required contextual information to the elementary face (=head) orientation information taken over from the ML/MF (*Freiwald et al., 2009*). This idea receives additional support from the fact that microstimulation of parietal area LIP causes stimulation-induced BOLD responses in a part of the STS whose coordinates seem to correspond to those of our study GF patch (*Crapse et al., 2013*). Area LIP is a

well-established center of overt and covert shifts of attention guided by a wide variety of cues, including head gaze (*Shepherd et al., 2009*; *Bisley et al., 2011*).

Using a comparable approach to delineate the cortical substrates of eye gaze following in humans, gaze following related BOLD activity was described bilaterally in the posterior STS ('pSTS region') (*Materna et al., 2008a*), later shown to be activated also by following other biological cues like finger pointing (*Materna et al., 2008b*) and head orientation (*Laube et al., 2011*). Importantly, presenting distracting eye gaze cue in conjunction with head gaze following led to a clear modulation of the BOLD signal in the pSTS (*Laube et al., 2011*). We have not studied if changing eye orientation relative to the head changes gaze-related activity in the GF patch in monkeys. This must not be expected in view of the fact that monkeys seem to pay very little attention to the other one's eye when trying to establish joint attention. However, independent of the differing weights attributed to eye and head gaze in humans and monkeys, the geometrical calculations needed to pinpoint a spatial goal shared by the two interacting agents are comparable. This consideration may suggest that the regions in the monkey and human STS activated by gaze following are analogous and possibly even homologous.

In contrast to the absence of overlap between the GF patch and the neighboring face patches, the BOLD activity related to the usage of facial identity for target localization in the anterior parts of the STS overlapped with activity evoked by the passive observation of faces. The identity-associated signal coincided with AM in M2 and AL in M1. The anterior location of our identity matching BOLD activation fits the previous electrophysiological finding of identity-selective neurons in the anterior inferotemporal gyrus (*Hasselmo et al., 1989*; *Eifuku et al., 2004*; *Freiwald and Tsao, 2010*). In accordance with the notion that these patches are involved in establishing facial identity, it was recently demonstrated that the percept of facial identity is disrupted by microstimulation of AM (and ML) face patches (*Moeller and Tsao, 2013*).

In summary, by requiring monkeys to use head gaze to locate spatial targets we could identify a highly specific region in the posterior STS (GF patch) well separated from those parts of the STS known to process visual information on faces and heads. We propose that this region corresponds to the pSTS of humans devoted to eye gaze following. We furthermore suggest that this region may be the substrate of the geometrical calculations needed to translate head orientation into precise shifts of attention.

## Materials and methods

### Subjects

Two rhesus monkeys (Macaca mulatta): monkey M1 (6 years, 8 kg); monkey M2 (8 years, 11 kg) were implanted with three cf-PEEK (carbon-fiber-enforced polyetheretherketone) tripods, each attached to the skull with six ceramic screws (Thomas Recording). Surgeries were carried out under combination anesthesia with isoflurane and remifentanyl with monitoring of all relevant vital parameters (body temperature, $CO_2$, blood oxygen saturation, blood pressure, ECG). After surgery, monkeys were supplied with opioid analgesics (buprenorphin) until they fully recovered. Every effort was made to minimize discomfort and suffering. The study was performed in strict accordance with the recommendations in the Guide for the Care and Use of Laboratory Animals of the National Institutes of Health. All of the animals were handled according to the guidelines of the German law regulating the usage of experimental animals and the protocols approved by the local institution in charge of experiments using animals (Regierungspraesidium Tuebingen, Abteilung Tierschutz, permit-number N1/08).

### Training and behavioral control

The monkeys were placed in an MR-compatible primate chair in a horizontal ('sphinx') posture. To immobilize the monkey's head, it was fixed to the chair by screwing the tripods to an acryl cap with an integrated massive cf-PEEK rod, which then was connected to the chair's frame. Before scanning they were trained in a mock scanner having the same geometry, projection screen and offering a similar acoustic environment as the real scanner. Hearing protection was guaranteed in the mock as well as in the real scanner by custom-made earmuffs with thick plastic foam filling. Habituating and subsequently training on the behavioral tasks took 6 months in M2 and 1 year in M1 until the mean accuracy scores in each task for the training session reached 80%. Eye position was tracked in real time using a scanner-compatible low-cost CMOS-infra-red camera (C-MOS-Kameramodul1(C-CAM-A), Conrad Elektronik, Germany) with infra-red emitting LEDs illuminating the monkey's eyes. The custom-made software

running on a standard PC determined orientation of the center of the pupil with a spatial resolution of 0.5° visual angle and a temporal resolution of 50 Hz. Camera and power supply cables were equipped with radio frequency (RF)-eliminating filters to minimize RF interactions in the real scanner (*Kamphuis et al., 2009*). Fluid rewards were supplied via a long and flexible tube, with the control unit, valve and reservoir installed outside the scanner room.

## Scanning

Monkeys were scanned in a horizontal clinical scanner, 3T MRT (Trio, Siemens, Erlangen, Germany). Functional images were acquired using two custom-made linear receiver surface coils (diameter 13 cm, 'Helmholtz-configuration'; whole-brain scanning in M1 and STS-ROI scanning in M2, later referred as 'bilateral coil') or one small surface coil (diameter 3 cm; STS-ROI scanning in M2 and additional left STS-ROI scanning in M1, later referred as 'focal coil'), placed above the temporal lobe (centered on the posterior STS). Each functional time series consisted of gradient-echoplanar whole-brain images (repetition time (TR) = 2000 ms; echo time (TE) = 28 ms; flip angle = 70°; 64 × 64 matrix; 1.31 × 2.42 × 1.31 mm voxels; 22 horizontal slices) or STS-ROI images (repetition time (TR) = 1000 ms; echo time (TE) = 28 ms: flip angle = 70°; 64 × 64 matrix; 1.31 × 2.42 × 1.31 mm voxels; 11 horizontal slices). For the purpose of the first experiment ('gaze following' paradigm) 10,364 whole brain volumes in 80 functional runs (see 'Visual stimuli' section for definition of a single functional run) were scanned in six scanning sessions (separate days of measurement) in monkey M1, using the bilateral coil. Additionally, 4400 STS volumes were scanned (55 functional runs, 2 scanning sessions) using the focal coil, centered on the left posterior STS in M1. In M2 a total of 12,280 right STS volumes were scanned in 75 functional runs: 6276 vol in 36 functional runs during 3 scanning sessions using the focal coil and 6004 in 39 functional runs during 3 scanning sessions with the bilateral coil. A total of 10,501 left STS volumes were scanned in 67 functional runs (4497 vol in 28 functional runs during 3 scanning sessions using the focal coil; 6004 vol in 39 functional runs during 3 scanning sessions with the bilateral coil). In the second experiment ('passive face perception' paradigm) a total of 4626 vol were scanned in 28 functional runs (3 scanning sessions) in M1 using whole-brain volumes and the bilateral coil. In M2 a total of 6106 STS volumes in 29 functional runs during 3 scanning sessions were collected from the right hemisphere (2724 vol in 13 functional runs during 1 scanning session using the focal coil; 3382 vol in 16 functional runs during 2 scanning sessions using the bilateral coil) and a total of 6297 STS volumes in 30 functional runs during 3 scanning sessions on the left hemisphere (2915 vol in 14 functional runs during 1 scanning session using the focal coil; 3382 vol in 16 functional runs during 2 scanning sessions using the bilateral coil).

## Visual stimuli

### 'Gaze following' paradigm (Experiment 1)

We used color images of demonstrator monkey faces (the largest height and width 5.6 × 5.6°) presented in the center, together with four targets (red dots, diameter 0.8°) drawn on a virtual horizontal line at −10°, −5°, 5° and 10° eccentricity as seen by the observer. The target eccentricities as seen by the demonstrator monkey were four times as large (i.e., −40°, −20°, +20°, +40°), reflecting the fact that the target plane was four times closer to the demonstrator than to the observer chosen to be closer to the demonstrator in order to demand large gaze shifts on his side (*Figure 1A*). We used photographs of different monkeys living in the same colony as the observers, taken while the individuals were sitting in the primate chair (head-fixed) with their head (and eyes) directed at a spatially well-defined object of attention. The raw images were processed using 'PAINT.NET' free software to erase the headholder and recording chambers from the portraits. They were mirrored horizontally in order to generate opposite head gaze directions. The stimuli were presented using an LCD projector (NEC GT 950, 1024 × 768 pixels) placed outside the scanner room. Images (32 × 24° visual angle) were back-projected via a 45° angled mirror on a translucent screen, inside the scanner bore at a distance of 60 cm from the monkey's eyes. Stimuli were presented in blocks consisting of 10–12 'orientation' trials (observer had to shift attention overtly), or 5–6 'fixation-only' trials. Blocks of 'fixation-only' trials were alternated with 'orientation' blocks based on gaze following (gf) or identity matching (im) task. Every functional run started with a 'fixation-only' block and contained two repetitions of each of two 'orientation' blocks (*Figure 1C*) while the order of gf and im blocks was pseudo-randomized across functional runs. Each trial started with the presentation of the portrait of a monkey oriented straight ahead, not part of the group of four portraits used to shift attention, with a small fixation target on the

portrayed monkey's forehead. The observer had to keep his eyes within a 4 × 4° window centered on the fixation target. 400 ms later, one of the portrayed monkeys, with its head turned to one of the four peripheral targets, appeared. Another 1200 ms later the fixation target turned off, telling the observer to make a saccade to one of the peripheral targets. The color and the shape of the central fixation cue told him which rule to apply in order to identify the correct peripheral target. In the case of a red circular fixation cue (0.8° diameter) the observer was required to saccade to the target the demonstrator was looking at (gaze following). In the case of a green rectangle (0.5 × 0.8°), the saccade target was identified by the learned association between the target locations and the four individual demonstrator monkeys whose portraits were shown (identity matching). As the portrait of each individual monkey could be shown in four different head gaze orientations, corresponding to the four target locations, the stimulus set involved 16 stimuli (*Figure 1A*) and the one used in a particular trial was chosen at random. Finally, in 'fixation-only' trials, indicated by a blue circle (0.8° diameter) fixation target, the observer had to withhold any eye movements and stay on the fixation target location within the fixation window until the end of the trial. In the case of 'orientation' trials, the fixation window was removed for 0.5 s after the go signal to give the observer the opportunity to move his eyes to the new target. Thereafter it was reestablished at the new target location for 1.3 s. A juice reward was delivered at the end of the trial if the observer had satisfied the task requirements (*Figure 1B*). Stimulus presentation and recording of eye movement data was controlled using an open source recording and stimulation system (nrec.neurologie.uni-tuebingen.de/nrec).

To assure that the observers actually used geometrical head gaze following in gaze following trials rather than resorting to potentially learned associations between head orientation as seen in the portrait and particular targets, we carried out two psychophysical control experiments outside the scanner (*Figure 2*). Like the standard experiment described before also the control experiments consisted of gaze following, identity matching and 'fixation-only' trials, however with interspersed catch trials. The first control experiment was designed to exclude learned associations between head orientation and spatial position of targets: There, in 6–12% of the gaze following trials (catch trials, *Figure 2A*, [II]) intermingled randomly with regular gaze following trials (*Figure 2A*, [I]), the stimulus monkey's portrait was shifted horizontally (shifts of −10°, −5°, 5° and 10°, with respect to the default central position (*Figure 2A*, [II]). The observer was asked to perform a saccade towards one out of four target positions after the offset of the fixation cue that had prompted gaze following. A reward was given on 50% of the catch-trials, independent of the target chosen, provided the observer stayed on the chosen target for at least 1 s. A total of 28 (M1) and 17 (M2) experimental sessions were performed, yielding altogether 460 (M1) and 260 (M2) catch trials respectively. The second control experiment tested for associations between head orientation and the ordinal position of targets within the sequence of four. Here, in the catch trials, the outer targets stayed in their standard locations of −10° and 10° respectively, whereas inner targets (normally at −5° and 5° respectively) were shifted further out to −15° and 15° eccentricity respectively. Thus, the 10° eccentricity targets maintained their standard spatial position but changed their ordinal position (from 1 and 4 to 2 and 3), (*Figure 2B*, [II]). All other aspects of this experiment corresponded to the ones given for control Experiment 1. A total of 11 (M1) and 12 (M2) experimental sessions were performed, yielding 50 (M1) and 51 (M2) catch trials respectively. Finally, to identify the strategies our monkey observers used in order to solve the identity-matching task, we repeated both control experiments outside the scanner with catch trials interspersed in identity matching trials (*Figure 2—figure supplements 1–4*). To this end the same procedure as described before for the control experiments related to the gaze following task were used. It was based on introducing catch trials interspersed in a regular identity-matching task. The catch trials were the same as those used in the control experiments testing for spatial associations and order associations in the gaze following task. A total number of 10 (M1) and 10 (M2) experimental repetitions were performed in the experiment testing for spatial associations with 150 (M1) and 70 (M2) catch trials. A total number of 10 (M1) and 12 (M2) experimental repetitions were performed in the control experiment testing for order associations with 150 (M1) and 50 (M2) catch trials.

## 'Passive face perception' paradigm (Experiment 2)

We used a set of different categories of black and white pictures: faces (monkey faces, human faces) and non-face objects (human bodies without head visible, human-made tools, fruits, hands). A second set of images was generated by scrambling the original ones (Adobe Photoshop CS5, scramble filter, 20 × 20 randomly shuffled blocks, [*Figure 4—figure supplement 1*]). The human faces were taken

from the Nottingham Scans database (free for research use under the terms of a Creative Commons Attribution license, http://pics.psych.stir.ac.uk). All other images were from a variety of freely available sources. They were chosen to be as similar as possible to the stimuli used in the previous studies (*Tsao et al., 2003*). The stimuli were presented using the same setup as the one used for the 'gaze following' paradigm. Each image had a size of 11 × 11°, was presented for 1s on a black and white random dot background (pixel size 0.05°) and was repeated once in each functional run. The monkey was rewarded for keeping his eye gaze within a fixation window of (2 × 2°) centered on a central fixation cue dot (0.5° diameter). Short breaks of fixation (not longer than 100 ms, mostly associated with eye blinks) were tolerated. Stimuli were presented in blocks of 16 images, all chosen randomly from the category of face stimuli or, alternatively, the various categories of non-face stimuli. Blocks of face stimuli (monkey faces or human faces) alternated pseudo-randomly with blocks of non-face objects (fruits, tools or headless bodies or hands). Each of these blocks was preceded by a block consisting of the scrambled versions of the following block. In each functional run, the sequence of 'scrambled faces, faces, scrambled non-faces, non-faces' was repeated four times (in total 16 blocks and 256 images). The serial position of the category (faces, non-faces) within the sequence was balanced across all functional runs.

## Data analysis

### 'Gaze following' paradigm (Experiment 1)

Eye movements records were analyzed offline (*Figure 1C*) in order to assess task performance, defined as the percentage of correctly chosen targets, in both gaze following and identity matching task. Only functional runs with success rates exceeding 70% in the two tasks were considered for further BOLD fMRI analysis. The hypothesis of a significant difference in accuracy between tasks was evaluated by running a Wilcoxon signed rank test (a Kolmogorov–Smirnov test had shown that the data were not distributed normally; p=0.001 [M1], p=0.001 [M2]). Response times (RTs) were calculated as the time between cue offset and the onset of the monkey's first saccade, the latter defined by an eye velocity threshold (>50°/s). Significant differences in RTs between the two tasks were detected with a paired *t* test. A Kolmogorov–Smirnov test did not show deviation from normality of 'gaze following': (M1: p=0.08, M2: p=0.06) and 'identity matching' (M1: p=0.2, M2: p=0.06) distributions. In order to test behavioral performance of M1 for gaze following to the left and to the right, we calculated separately the response accuracy to demonstrator's left and right gaze for each gaze following block (138 in total). As the Kolomogorov–Smirnov test had shown that the two data sets were not distributed normally (p=0.001), we used related-samples Wilcoxon signed rank test to assess the significance of the difference between the median response accuracies to the right (Median = 100%, 95% CI = 100–100%, n = 138) and to the left (Median = 100%, 95% CI = 93.07–106.93%, n = 138) sites. The difference was not significant (Z = 0.15, p=0.88). To test for spatial and/or order associations of the subject solving the gaze following task, we examined a subject's responses to catch trials in the two control experiments. The response was defined as the first horizontal saccade to one out of four targets after the offset of the central cue, followed by fixation on the target for at least 1 s. The responses were classified into 3 categories. In the first control experiment, testing for spatial associations, these were the 'gaze following' category (the chosen target position was congruent with gaze direction of the stimulus monkey's portrait, now shifted horizontally together with the stimulus monkey's portrait), the 'learned spatial association' category (the chosen target position corresponding to the correct one before the horizontal shift of the appropriate monkey's portrait, indicating that the monkey exploited a learned association between the given portrait and the absolute target position in space) and the 'other' category (the subject's response not compatible with any of the previous categories), (*Figure 2A*, [III]). In the second control experiment testing for ordinal associations we defined the following response categories: the 'gaze following' category (the chosen target position was congruent with the gaze direction of the portrayed monkey, irrespective of the changed ordinal position), the 'learned order association' category (the target shifted further out, i.e., the one with the largest/smallest ordinal position, is preferred, although gaze is directed at the inner target positions, indicating that the monkey relies on learned associations between portraits and target ordinal position) and the 'other' category (the subject's response not compatible with any of the previous categories), (*Figure 3B*, [III]). We used a repeated measures one-way ANOVA to compare the differences in the percentages of correct responses across the strategies for each subject (Kolmogorov–Smirnov test first confirmed that the data were normally distributed: p>0.05 in M1 and M2).

To test for strategies used in order to solve the identity-matching task we classified the subject's responses in catch trials interspersed in a regular identity-matching task into three strategy categories. In the experiment testing for spatial associations (*Figure 2—figure supplement 1*), those were: 'learned absolute spatial associations' category (target chosen according to its absolute spatial position associated with a given demonstrator's facial identity), 'learned relative spatial associations' category (target chosen according to its relative distance with respect to a given demonstrator's portrait, associated with its facial identity) and 'other' category. In the experiment testing for ordinal associations (*Figure 2—figure supplement 2*), we identified: 'learned absolute spatial associations' category (target chosen according to its absolute spatial position associated with a given demonstrator portrait's identity), 'learned order associations' category (target chosen according to its ordinal position in the target order associated with a given demonstrator's identity) and 'other' category.

Functional images of both monkeys were analyzed using the statistical parametric mapping program package SPM8 (Wellcome, Department of Cognitive Neurology, London, UK, http://www.fil.ion.ucl.ac.uk/spm) implemented in Matlab 7.9 (R2009b). Images were spatially aligned, co-registered with the mean functional image which in turn was co-registered with the structural image of the individual monkey and finally spatially smoothed with a Gaussian filter (3 mm full-width-half-maximum). We calculated response levels (β values) associated with each experimental task for each voxel using a statistical analysis based on the general linear model (GLM). The BOLD response during each experimental task block was modeled as a boxed-car covariate of variable length in SPM8 using a human canonical hemodynamic function. Assuming a standard human hemodynamic function seemed pragmatic as previous work has not suggested fundamentally different hemodynamic responses in monkeys and humans (for instance compare the similar BOLD responses of both species in auditory (*Bauman et al., 2010*) and visual cortex (*Boynton et al., 1996*; *Logothetis et al., 2001*; *Logothetis, 2002*)). Regressors representing the estimated head movements (translation and rotation; altogether six degrees of freedom) were added to the model as covariates of no interest to account for artifacts due to head movements during scanning. Contrast analysis comparing the gaze following and the identity matching conditions were carried out for both monkeys. Significant changes were assessed using t-statistics. In order to take the large number of data from each monkey into account, we used a fixed effect model to analyze each successful scanning day individually (a total of 6 days for M1, a total of 6 days for M2) and then analyzed the contrast images provided by each model using a second-level random effects analysis. We labeled a contrast as significant if a single-voxel threshold of p<0.005 (uncorrected) was met in at least five contiguous voxels. To analyze the data collected in the 'focal coil' experiments in M1 experiments we were able to perform fixed effect analysis based on the complete data pool. In this case, two different statistical significance levels were compared, a single-voxel threshold of p<0.005 (uncorrected) met in at least five contiguous voxels or, alternatively, a single-voxel threshold of p<0.05 (uncorrected), again in at least five contiguous voxels (*Figure 3—figure supplement 1*). For the purpose of visualization we used Caret (http://brainvis.wustl.edu/wiki/index.php/Caret:About), which offered a flattened reconstruction of the cortical surface gray matter onto which the statistical t-map was projected.

## 'Passive face perception' paradigm (Experiment 2)

Eye movement data were analyzed to assess the accuracy of the fixation. Functional runs in which the monkey failed to stay inside the fixation window of (2 × 2°) in at least 85% of the trials were rejected. The preprocessing of the MRI data followed the procedure described above, the only difference was the size of the full-width-half-maximum of the Gaussian filter used for spatial smoothing (here 2 mm). To define face-selective regions, we calculated the contrast 'faces vs other objects' (not considering scrambled images). The output was masked with a contrast of 'both faces and other objects vs all their scrambled counterparts' (thresholded to p<0.05, uncorrected) to identify voxels selective only for complex images rather than for simple visual patterns. This procedure was in accordance with the one described previously (*Tsao et al., 2003*, *2008*). Because of the smaller amount of data collected here in comparison with the 'gaze following' paradigm (Experiment 1), we performed a fixed-effect analysis pooling all data obtained for each subject monkey. Statistical significance was assumed if a single-voxel threshold of p<0.005 (uncorrected) in at least five contiguous voxels (or p<0.05 uncorrected, 5 contiguous voxels) was met.

## Additional information

### Funding

| Funder | Grant reference number | Author |
|---|---|---|
| Deutsche Forschungsgemeinschaft | TH 425/12-1 | Peter Thier |

The funder had no role in study design, data collection and interpretation, or the decision to submit the work for publication.

### Author contributions

KM, Conception and design, Acquisition of data, Analysis and interpretation of data, Drafting or revising the article; AA, PWD, Acquisition of data, Drafting or revising the article; PT, Conception and design, Drafting or revising the article

### Ethics

Animal experimentation: This study was performed in strict accordance with the recommendations in the Guide for the Care and Use of Laboratory Animals of the National Institutes of Health. All of the animals were handled according to the guidelines of the German law regulating the usage of experimental animals and the protocols approved by the local institution in charge of experiments using animals (Regierungspraesidium Tuebingen, Abteilung Tierschutz, permit-number N1/08). All surgery was performed under combination anesthesia involving isoflurane and remifentanyl and every effort was made to minimize discomfort and suffering.

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
