## [Decision Letter]

Thank you for sending your work entitled “Disparate substrates for head gaze following and face perception in the monkey superior temporal sulcus” for consideration at *eLife.* Your article has been favorably evaluated by a Senior editor and 2 reviewers, one of whom is a member of our Board of Reviewing Editors.

The Reviewing editor and the other reviewers discussed their comments before we reached this decision, and the Reviewing editor has assembled the following comments to help you prepare a revised submission.

This study presents interesting fMRI data on the cerebral cortex region associated with the active head-gaze following in the behaving monkey. The main result is the finding of an activated region after subtraction of two tasks (active head-gaze-following and identity-matching) in the posterior STS that does not overlap the activated region of passive face recognition region. The results appear to clarify the neural mechanisms of gaze-following behavior. There are some points that the authors must address in a revised version of the manuscript:

1) The authors conclude that the gaze-following patch exists outside the passive face patch and that it could not be elicited directly. The authors should discuss this point. Discrepancy between the two monkeys that pSTS unilateral activity in one animal and bilateral activities in the other monkey could be argued from this point of view.

2) Previous studies done by the same group (26) demonstrated the active gaze-following exists in the middle STS but not in pSTS. The authors should discuss about differences between the two studies and should mention the study of Kamphsuis in the Discussion section. An obvious question is why the authors did not mention this previous study.

3) What cue the monkey used to choose one target out of 4 in the gaze-following test is a very difficult question. The study showed that the monkey used the gaze-direction, the face single angle of the monkey's photos as the cues. However, one problem to be solved still remains, the possibility that the monkey simply learned the association between the face angle and the order of the target from the face photo. The authors should discuss this possibility.

---

## [Author Response]

*1) The authors conclude that the gaze-following patch exists outside the passive face patch and that it could not be elicited directly. The authors should discuss this point. Discrepancy between the two monkeys that pSTS unilateral activity in one animal and bilateral activities in the other monkey could be argued from this point of view*.

As suggested, we added that the unilaterality of the gaze following-related activity in one of the monkeys vis-à-vis largely bilateral face-evoked activity is yet another finding supporting the idea that the gaze following patch is anatomically distinct from the face patches.

*2) Previous studies done by the same group (*[26]*) demonstrated the active gaze-following exists in the middle STS but not in pSTS. The authors should discuss about differences between the two studies and should mention the study of Kamphsuis in the Discussion section. An obvious question is why the authors did not mention this previous study*.

Actually, the Kamphuis et al study was mentioned in the previous version. However, we agree that the findings should have been compared in more detail. This deficiency is ironed out in the revised manuscript: the discussion now offers a paragraph on the comparison.

*3) What cue the monkey used to choose one target out of 4 in the gaze-following test is a very difficult question. The study showed that the monkey used the gaze-direction, the face single angle of the monkey's photos as the cues. However, one problem to be solved still remains, the possibility that the monkey simply learned the association between the face angle and the order of the target from the face photo. The authors should discuss this possibility*.

This is an important concern, which we tried to address with the behavioral control experiments. The results obtained clearly rule out that our experimental animals had relied on learned associations. In the revised manuscript this important result receives additional attention by having added a paragraph in the Discussion.